# Composition and Antibacterial Activity of the Essential Oil from *Pimenta dioica* (L.) Merr. from Guatemala

**DOI:** 10.3390/medicines7100059

**Published:** 2020-09-23

**Authors:** Max Samuel Mérida-Reyes, Manuel Alejandro Muñoz-Wug, Bessie Evelyn Oliva-Hernández, Isabel Cristina Gaitán-Fernández, Daniel Luiz Reis Simas, Antonio Jorge Ribeiro da Silva, Juan Francisco Pérez-Sabino

**Affiliations:** 1Escuela de Química, Facultad de Ciencias Químicas y Farmacia, Universidad de San Carlos de Guatemala, EdificioT-12, Ciudad Universitaria, zona 12, Guatemala City 01012, Guatemala; maxmerida2050@gmail.com (M.S.M.-R.); mwmanuel@yahoo.com (M.A.M.-W.); bessieoliva@yahoo.com (B.E.O.-H.); 2Escuela de Química Biológica, Facultad de Ciencias Químicas y Farmacia, Universidad de San Carlos de Guatemala, EdificioT-11, Ciudad Universitaria, zona 12, Guatemala City 01012, Guatemala; isabelgaitan1810@gmail.com; 3Instituto de Pesquisas de Produtos Naturais, Centro de Ciências da Saúde, Universidade Federal do Rio de Janeiro, Bloco H, Ilha do Fundão, Rio de Janeiro CEP 21941-590, Brazil; danielsimas16@gmail.com (D.L.R.S.); ajorge@ippn.ufrj.br (A.J.R.d.S.)

**Keywords:** eugenol, gas chromatography, mass spectrometry, traditional medicine

## Abstract

**Background:***Pimenta dioica* is a native tree of Central America, Southern Mexico, and the Caribbean used in traditional medicine. It grows in wet forests in the Guatemalan departments of Petén and Izabal. Since the plant is not being economically exploited in Guatemala, this study was aimed at determining the composition of the essential oil of *P. dioica* leaves and fruits and the antibacterial activity of the leaves in order to evaluate its possible use in health products. The essential oils of fruits and leaves are used as rubefacient, anti-inflammatory, carminative, antioxidant, and antiflatulent in different countries. **Methods:** Fruits and leaves of *P. dioica* from Izabal Department were collected in April 2014 and extracted by hydrodistillation method. The oils were analyzed by gas chromatography coupled with mass spectrometry (GC/MS). **Results:** Yields of 1.02 ± 0.11% for dried leaves and 1.51 ± 0.26% for fruits were obtained. Eugenol was the main component (65.9–71.4%). The leaf oil showed growth inhibition against two Gram-positive and two Gram-negative bacteria. **Conclusions:** The authors consider that the tree’s leaves can be evaluated as a source of ingredients for antiseptic products, and that it is important to evaluate other types of properties such as anti-inflammatory activity.

## 1. Introduction

*Pimenta dioica* (L.) Merr is a tree of the Myrtaceae family that grows to 20 m high and 30 cm in diameter, with aromatic leathery leaves 9 to 20 cm long. It is found in Southern Mexico, Belize, Central America, and the Antilles. In Guatemala it grows naturally in the departments of Petén and Alta Verapaz [1]. It produces a very aromatic fruit of 4–8 mm in diameter which is consumed as a spice. It is known in Guatemala as “fat pepper” and the English name “allspice” derives from the fact that it combines the flavors of cloves, cinnamon, nutmeg, and black pepper [1,2].

In Guatemala, the fruits of *P. dioica* are used to flavor food and in domestic medicine [1]. For example, the Mayans and Aztecs used them to embalm corpses due to their preservative qualities [3]; it is used to promote digestion, to expel stomach gases, as a mild anesthetic for gum, teeth, muscle pain, menstrual cramps, arthritis, fatigue [3,4]; and it is cooked and ingested in Cuba, to treat colds and stomach pain [5]. The leaves of *P. dioica* have been used to relieve dental and muscular and rheumatic pain, colds, menstrual cramps, indigestion, flatulence, and diabetes [4]; and in India, the leaves are used by natives as medicine for pain, arthritis, fever, and stress [6].

The composition of the essential oil of fruits of *P. dioica* from Jamaica showed presence of eugenol (68–78%), methyl eugenol (2.9–13%), caryophyllene (3.30–4.90%), 1,8-cineol, α- phellandrene, humulene, and terpinolene [3,7]; a study in Mexico reported eugenol (90%) and α-terpineol (2%) in the essential oil of fruits [8]; eugenol (71.7%), β-myrcene (11.2%), (*E*)-caryophyllene (7.7%) were previously reported in the essential oil of Guatemalan fruits [9]. The composition of the essential oil of *P. dioica* leaves from Jamaica showed presence of eugenol, α-pinene, caryophyllene, limonene, and 1,8-cineole [3]; while eugenol (85.33%), β-caryophyllene (4.36%), cineole (4.19%), linalool (0.83%) and α-humulene (0.76%) were present in the essential oil of *P. dioica* leaves from Sri Lanka [10]. Eugenol (76.20%), β-myrcene (10.70%), and (*E*)-caryophyllene (4.20%) were present in the essential oil from *P. dioica* leaves from Guatemala [9]; and a supercritical fluid extract of leaves of *P.dioica* from Mexico presented eugenol (77.90%) as the major component [11].

Different organs of *P. dioica* have shown positive biological activity in different pharmacology tests. The essential oil of the fruits of *P. dioica* from Jamaica showed high anti-radical activity against 2,2-diphenyl-1-picryl-hydrazyl-hydrate (DPPH) free radical and 2,2′-Azinobis-(3-ethylbenzothiazoline-6-sulfonic acid (ABTS) [7]. The tannin casuariin from the leaves of *P. dioica* of Egypt showed strong cytotoxic effect against hepatocellular carcinoma cells (Hep-G2), against human colon cancer cells (HCT-116), and mild cytotoxicity against human breast cancer cells (MCF-7) [12]. The aqueous suspension of the fruits of *P. dioica* showed antiulcer and cytoprotective activity of the gastric mucosa against indomethacin in rats [13]. The final aqueous fraction of the leaves of *P. dioica* from Costa Rica showed a hypotensive effect in spontaneously hypertensive rats [14]; and the ethanol extract of *P. dioica* leaves showed significant liver protection in Wistar rats poisoned with carbon tetrachloride [6].

The fruits of *P. dioica* have shown an effect against parasites of commercial importance. Nematicidal property of the essential oil of *P. dioica* from Jamaica against *Bursaphelenchus xylophilus* was confirmed [15] and the essential oil of the fruits showed an acaricidal effect against larvae of *Rhipicephalus microplus* [16].

The antimicrobial activity of extracts and essential oils of fruits and leaves of *P. dioica* has been demonstrated. *P. dioica* leaf extracts showed significant antimicrobial properties against *Staphylococcus aureus*, *Streptococcus mutans*, *Escherichia coli*, *Bacillus cereus*, *Salmonella typhimurium*, *Candida albicans*, *Pseudomonas fluorescens*, *Bacillus megaterium*, *Aspergillus niger,* and *Penicillium* sp. [17,18,19,20]. In another study the essential oil of *P. dioica* leaves showed growth inhibition of *Staphylococcus aureus*, *Escherichia coli*, *Salmonella typhi*, *Pseudomonas aeruginosa,* and *Bacillus cereus* [21]. The essential oil of *P. dioica* fruits showed significant antimicrobial properties against *Acinetobacter baumannii*, *Staphylococcus aureus*, *Pseudomonas aeruginosa*, *Candida albicans,* and other yeasts of the *Candida* genus [8,22].

The purpose of the present study was to determine the composition and the antibacterial activity of *P. dioica* essential oil from the northeastern region of Guatemala, evaluating the activity against Gram-positive and Gram-negative bacteria, in order to assess its potential as a broad-spectrum natural antiseptic.

## 2. Materials and Methods

### 2.1. Plant Material

The leaves and fruits of *P. dioica* were collected from six trees selected at random with the aid of Q’eqchí people in April 2014 in the community of Barra Lampara, Livingston, Izabal Department at an altitude of 13 meters above sea level (masl). A voucher sample was deposited in the Herbarium of the School of Biology (BIGU) of the University of San Carlos with code BIGU 72838. Collected plant material was separated into leaves and fruits, the they were dried in a soler dryer at temperature between 28 and 32 °C. Leaves and fruits were milled and extracted separately.

### 2.2. Extraction of Essential Oil

Oils of dried fruits and dried leaves of *P. dioica* were extracted in triplicates from 50.0 g of each organ by hydrodistillation method for 2 h in a Clevenger-type apparatus for 2 h and collected in pentane. After extraction, pentane was removed with a rotatory evaporator at 40 °C and the oils were weighed in an analytical scale, yielding 1.37% (*w*/*w*) for fruits and 0.58% (*w*/*w*) for leaves. All the extractions were made in triplicate and the reported yield corresponds to the average of the three extractions plus–minus a standard deviation.

### 2.3. Gas Chromatography Coupled to Mass Spectrometry Analysis (GC/MS)

Composition analyses were performed by GC/MS using a chromatograph Shimadzu 2010 Plus system fitted with a DB5-MS capillary fused silica column (60 m, 0.25 mm I.D., 0.25 µm film thickness) and coupled with a SHIMADZU QP-2010 Plus selective detector (MSD). Compound identification was based on the comparison of their mass spectra with the Standard Reference Database National Institute of Standards and Technology (NIST) 11 and of their retention indices with those compiled by Adams [23]. Quantitation was made on the basis of their chromatographic peak area percentages.

The initial oven temperature was set at 60 °C, then raised by 3 °C/min to 246 °C and held for 20 min. He (99.999%) was used as the carrier gas with a flow rate of 1.03 mL/min; the injector temperature was set to 220 °C and a split ratio of 1:50 was used. The oils were diluted in chloroform (5 µL oil in 450 µL chloroform) and 1 µL of the dilution was injected in the chromatograph. No replicates were performed. Mass spectra were taken at 70 eV. The range values used were of 40–700 *m*/*z*.

### 2.4. Antibacterial Activity

The antimicrobial activity assay of the essential oil of *P. dioica* leaves was performed using the agar diffusion method recommended by the National Committee for Clinical Laboratory Standards (NCCLS) (Kirby–Bauer method). The assay disks (Whatman, 6 mm, 420 g/m^2^) were impregnated with 10 uL of pure essential oil and allowed to dry for 24 h. A Mueller Hinton agar was inoculated using a bacteria concentration equivalent to the MacFarland 0.5 standard, in three directions and the oil disks were arranged at equivalent distances. The boxes were incubated at 37 °C or 24 h to subsequently measure the inhibition halo in millimeters. The assays for the oils with each bacteria were carried out in triplicate [24,25,26,27,28].

## 3. Results

### 3.1. Chemical Composition of the Essential Oil of P. dioica

Extraction yield of the essential oil was 1.02 ± 0.11% (*w*/*w*) for dried leaves and 1.51 ± 0.26% (*w*/*w*) for dried fruits. Refractive indices of the oils were determined at 20 °C as part of the characterization (1.5260 for leaves and 1.5255 for fruits). Thirty-five compounds had chromatographic area percentages greater than 0.1% in at least one oil (Table 1). Twenty-two compounds were identified in the two oils; the 22 compounds were present in the fruit oil and 21 (six as traces) in the leaf oil. The major component of both oils was eugenol with 71.4% for leaves and 65.9% for fruits (Table 1), β-myrcene was the second major component in leaves (10.0%) and fruits (10.1%). (*E*)-caryophyllene was the third major component (5.2% in leaves and 9.1% in fruits). In this study, the components were only quantified in percentages greater than 0.5%, since what was sought was to relate the main components to antibacterial activity.

### 3.2. Antibacterial Activity

Antibacterial activity of the essential oil of leaves of *P. dioica* was found against the four bacteria tested: two Gram-positive bacteria (*Staphylococcus aureus* and *Bacillus subtilis*) and two Gram-negative (*Escherichia coli* and *Salmonella enterica* serovar *Tiphymurium*). The highest activity was found against *B. subtlis* (inhibition zone: 23 mm) meanwhile the oils showed the lowest activity against *E. coli* (inhibition zone: 16 mm).

## 4. Discussions

The essential oil of dried leaves and dried fruits of *P. dioica* from Guatemala presented yields of (1.02 ± 0.11)% (*w*/*w*) and (1.51 ± 0.26)% (*w*/*w*) respectively and eugenol (71.4% in leaves and 65.9% in fruits) as the major component, and β-myrcene and (*E*)-caryophyllene in significant percentages (chromatograms of the oils of leaves and fruits are presented in Appendix A respectively, and the chromatographic reports in Appendix A for leaves, and Appendix A for fruits), in a comparable way to results previously found for the same species in Guatemala and other countries [3,7,8,9,10]. This indicates that the oil has low variability in terms of its chemical composition. *P. dioica* grows in Guatemala in two northern departments inhabited by Mayan people, especially the Q’eqchi’ ethnic group characterized by high poverty indices, who use the aerial parts in traditional medicine mainly as anesthetic and as a digestive, without taking advantage of it economically [1,2,3].

The oil of the leaves showed growth inhibition against the four bacteria tested (Table 2), showing the highest activity against *B. subtilis* (23 ± 1.2 mm inhibition zone) and the lowest against *E. coli* (16 ± 1.0 mm inhibition zone). Photographs of three tests are presented in Figure 1. The activity of the oil against these bacteria could be related to the synthesis of the cell wall, mainly interfering in the formation of the peptidoglycan molecule, which corresponds to up to 50% of the total cell wall of Gram-positive bacteria and in a lower percentage of Gram-negative [8,28].

The defective synthesis of the cell wall can be reflected in the loss of the selectivity of the cell and cause an osmotic imbalance, favoring bacterial lysis. Another possible mechanism of action is interference with protein synthesis in the 30s and 50s subunits of the bacterial ribosome, a mechanism that would make *P. dioica* an oil with a broader range of activity against other bacteria [24,28].

The authors suggest that this activity can be attributed to eugenol and (*E*)-caryophyllene since these compounds have been found recurrently in the essential oils of *P. dioica* leaves [3,9,10,11]. In essential oils that have a high content of eugenol, this compound has been attributed as the main one responsible for the antimicrobial activity, as in the case of clove essential oil [29]. Among the mechanisms that have been proposed for the action of eugenol against bacteria, a recent review about the antimicrobial activity of eugenol mentions that eugenol has a disruptive action on the cytoplasmic membrane and that it inhibits the production of virulence factors such as violacein, elastase, and procyanin. In the case of *E. coli*, eugenol acts by reducing the production of pyocyanin, inhibits the formation of enterohemorrhagic *E. coli* biofilms, as well as the mechanisms responsible for activity. In the case of *S. aureus*, eugenol inhibits and eradicates the biofilms produced by this bacterium [30].

The strains used in the study are certified strains that are known to be susceptible to different antimicrobials, as they are used in the quality control of antibiogram assays to measure the susceptibility of wild strains isolated from patients, therefore no positive control was required [24,28].

The essential oil of the fruit of *P. dioica* is of economical relevance in other countries as a spice for food seasoning [1,2,3]. The authors consider that *P. dioica* leaves could be sustainable used for the production of essential oil as an alternative for the development of antiseptic products and improve the income of Q´eqchi´ people, since the plant has abundant foliage and the oil has an acceptable extraction yield. Future studies should also be carried out regarding the anti-inflammatory activity of the essential oil of *P. dioica,* especially of the leaves.

## Figures and Tables

**Figure 1 medicines-07-00059-f001:**
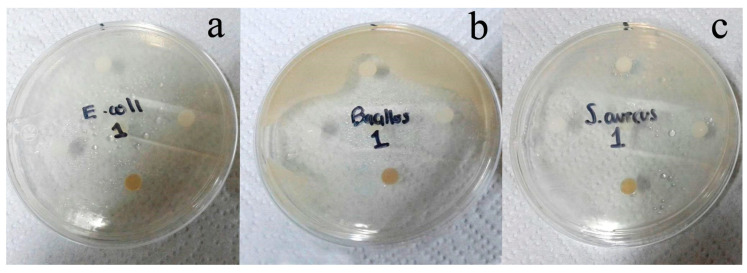
Antibacterial tests of the essential oil of leaves of *P. dioica* against (**a**) *E. coli*, (**b**) *B. subtilis,* and (**c**) *S. aureus.*

**Table 1 medicines-07-00059-t001:** Composition of the essential oil of fruits and leaves of *P. dioica* determined by gas chromatography coupled to mass spectrometry analysis (GC/MS) and Retention Index (KI).

Compound Id (MS)	SI	KI exp	KI Adams	% (Fruits)	% (Leaves)
β-myrcene	96	991.9	991	10.1	10.0
α-terpinene	94	1018.0	1017	0.1	t
o-cymene	93	1025.2	1026	0.1	0.1
limonene	95	1029.6	1029	0.4	0.4
1,8-cineole	97	1032.8	1031	0.8	1.6
(*E*)-β-ocimene	96	1045.7	1050	1.3	1.8
γ-terpinene	95	1057.9	1060	0.2	t
terpinolene	95	1085.9	1089	0.3	0.2
linalool	96	1101.5	1097	1.6	0.8
Terpinen-4-ol	96	1180.8	1177	1.0	0.6
α-terpineol	95	1195.4	1189	0.1	0.2
chavicol	96	1251.4	1250	0.8	3.1
eugenol	97	1363.7	1359	65.9	71.4
NI	---	1368.1	1369	0.1	nd
α-ylangene	94	1375.6	1375	1.2	t
NI	---	1388.6	---	t	0.2
methyleugenol	89	1401.7	1404	1.9	0.1
(*E*)-caryophyllene	95	1418.8	1419	9.1	5.2
α-humulene	95	1452.3	1455	1.2	1.1
γ-muurolene	92	1472.1	1480	0.2	t
germacrene D	93	1477.4	1485	0.4	t
NI	---	1487.1	--	0.1	0.5
NI	---	1491.6	---	0.2	0.6
NI	---	1509.3	---	0.1	t
δ-amorphene	90	1515.3	1512	0.8	t
caryophyllene oxide	91	1576.0	1583	0.4	0.7
NI	---	1611.8	---	nd	0.2
NI	---	1650.3	---	0.1	0.7
NI	---	1943.8	---	0.3	nd
NI	---	1978.8	---	0.1	nd
				98.9	99.5

SI: Similarity Index (Standard Reference database National Institute of Standards and Technology (NIST) 11); KI: Retention Index; NI: not identified; nd: not detected; t: traces; (---): No correspondence of KI with the MS structure.

**Table 2 medicines-07-00059-t002:** Antibacterial activity of the essential oil of *P. dioica* leaves.

Strain	Inhibition zone (mm) (Average ± Standard Deviation)
*Escherichia coli* ATCC *8739*	16 ± 1.0
*Bacillus subtilis* ATCC *6633*	23 ± 1.2
*Staphylococcus aureus* ATCC 25923	20 ± 0.6
*Salmonella enterica Tiphymurium*	22 ± 0.6

Source: Experimental results.

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
