# Peer review of "Composition and Antibacterial Activity of the Essential Oil from *Pimenta dioica* (L.) Merr. from Guatemala"

_medicines, 2020, doi:10.3390/medicines7100059_

Round 1

Reviewer 1 Report

This article is very interesting for anti-inflammatory activity of the essential oil of P. dioica, especially concerning the leaves.

The methodology is well described, but it was missed:

-in what period (month) of the year the leaves and fruits were harvested;

- have the leaves been distilled fresh or dried?;

- which rule was chosen for the collection of leaves and fruits (at random?).

Author Response

The month and the type of collection were described in the methodology (lines 101-107) and that the leaves and fruits were dried before extraction (109-112). English also was checked and corrected throughout the text (Track changes function was used).

Author Response

The methodology was improved. It was explained that the extraction (109) and antibacterial activity (line 139, not changed) were performed in triplicate. The GC/MS was only one measurement (lines 127-128) and it is not possible to repeat the analysis since the sample was used for antibacterial activity assays and other tests.

A similarity index “SI” column was added to Table 2 (lines 153-155). The information of the database was provided in the methodology (lines 119-120) and below the table (line 155) and other minor changes were made in the legend and columns to be more explicative. 

The information about previous identification of substances in the oil P. dioica is in the Introduction (lines 60-71) so we did not repeat it in the table 1. A figure with photographs of three antibacterial tests was included using the best picture available (lines 152-154. References that support the discussion were placed in the corresponding paragraphs (lines 175, 181, 190, 192, 194, 201, 205- 207).

The English was checked and corrections were made regarding the typographic and grammar mistakes that we found throughout the text (“Track Changes” function was used).

Round 2

Reviewer 2 Report

The authors addressed all my concerns.